# Recent Advances and Future Potential of Long Non-Coding RNAs in Insects

**DOI:** 10.3390/ijms24032605

**Published:** 2023-01-30

**Authors:** Junaid Zafar, Junlin Huang, Xiaoxia Xu, Fengliang Jin

**Affiliations:** Key Laboratory of Bio-Pesticide Innovation and Application of Guangdong Province, College of Plant Protection, South China Agricultural University, Guangzhou 510642, China

**Keywords:** long non-coding RNAs, miRNAs, insects, invertebrates, lncRNAs

## Abstract

Over the last decade, long non-coding RNAs (lncRNAs) have witnessed a steep rise in interest amongst the scientific community. Because of their functional significance in several biological processes, i.e., alternative splicing, epigenetics, cell cycle, dosage compensation, and gene expression regulation, lncRNAs have transformed our understanding of RNA’s regulatory potential. However, most knowledge concerning lncRNAs comes from mammals, and our understanding of the potential role of lncRNAs amongst insects remains unclear. Technological advances such as RNA-seq have enabled entomologists to profile several hundred lncRNAs in insect species, although few are functionally studied. This article will review experimentally validated lncRNAs from different insects and the lncRNAs identified via bioinformatic tools. Lastly, we will discuss the existing research challenges and the future of lncRNAs in insects.

## 1. Introduction

With the inception of high-throughput genomic technologies, including microarray and next-generation sequencing (NGS), researchers have detected several RNA transcripts in particular tissues or cells at a precise time. Apart from the conventional messenger RNAs (mRNAs) that translate into functional proteins, RNA-sequencing (RNA-seq) has put the spotlight on various classes of non-coding RNAs (ncRNAs) [1]. Once believed to be transcriptional noise, junk RNAs, or the by-products of genetic operations, ncRNAs have emerged as promising biomarkers [2]. The ncRNAs comprise the largest class of RNAs and are classified into small ncRNAs (sncRNAs) and long ncRNAs (lncRNAs) [3]. In the past decade, lncRNAs have achieved global interest due to their potential role in numerous biological processes, including alternative splicing, cell cycle, epigenetics, dosage compensation, and gene expression regulation [4,5].

Even though lncRNAs have been documented by several taxonomic groups, and many critical discoveries are made in plants and mammals [6,7], our discussion will mainly focus on insects, with occasional references to other species.

## 2. Historical Overview

In the 1980s, the intricate mechanism of genomic imprinting was discovered, an epigenetic process that determines gene expression based on whether they are inherited from the father or the mother. Two imprinted genes: *Igf2r* (encoding the receptor for insulin-like growth factor type-2), the paternally expressed protein-coding gene, and the maternally expressed *H19,* were identified in mice. These genes located on chromosome 7 in close proximity, formed the H19/IGF2 cluster [8,9]. However, the surprising lack of translation in *H19* continued to puzzle the researchers until a functional description of an additional lncRNA, *Xist* (X-inactive-specific transcript), involved in dosage compensation among mammals [10,11] was performed. Based on this information, *H19* can be considered the first eukaryotic non-coding gene ever discovered; however, it was initially described as mRNA [12]. The ground-breaking discoveries of *H19* and *Xist* transformed our perspective of ncRNAs and their functional relevance. 

However, as the traditional biological methods were not powerful enough, the research concerning lncRNAs remained steady until the dawn of the 21^st^ century, when the FANTOM project revealed several putative lncRNA transcripts by employing full-length cDNA cloning [13]. This evidence rejuvenated the scientific community’s interest, which led to the development of several methods describing the lncRNAs, including the Tiling array and ChIP-Seq [14,15,16]. Figure 1 illustrates some of the significant discoveries made in the field of lncRNAs. Despite these technological advances, rapid progress in insects was made after the development of RNA-seq [17], and several lncRNAs were discovered and cataloged from the *Drosophila melanogaster* genome [18].

## 3. Biogenesis and Classification

Although lncRNAs are classified as ncRNAs, they are distinctly different from other types of ncRNAs; hence, it is essential to understand their biogenesis for functional significance. LncRNAs are described as RNA molecules with a length of ≥200 nucleotides (nt) that lack protein-coding ability [19,20]. Similar to mRNAs, lncRNAs are transcribed by RNA polymerase (RNAP) II/III, spliced and capped at the 5` [21], polyadenylated, and contain exons (which are longer but fewer in contrast to mRNAs) [22]. 

Researchers have established that different types of lncRNAs are transcribed from numerous DNA elements such as promoters, enhancers, and intergenic regions [23]. Unlike other classes of ncRNAs, lncRNAs benefit from secondary and 3D structures, enabling them to have RNA and protein-like functions [24]. The lncRNAs are more highly enriched in the nucleus than in the cytoplasm. For instance, in humans, 17% of lncRNAs are enriched in the nucleus compared to 4% in the cytoplasm [25]. However, multiple lncRNAs have been detected that are shuttled into the cytoplasm after synthesis in the nucleus and mediate numerous modes of gene regulation [26,27]. The cytoplasmic lncRNAs play crucial roles, including mRNA translation regulation, mRNA stability, microRNA (miRNA) precursors, and miRNA sponges/decoys by sequestering miRNAs and reducing their regulatory effects on target mRNAs [28,29,30,31]. In addition, lncRNAs are released into extracellular space as circulating lncRNAs, which could potentially be biomarkers [32]. The biogenesis of lncRNAs is very complicated, and additional in-depth investigations are required for a better understanding [33,34]. Even though lncRNAs and mRNAs share various characteristics, several types of lncRNAs are unique in biogenesis, form, and function [35]. Unlike mRNAs, lncRNAs are expressed in a highly spatiotemporal pattern and at substantially lower levels. They commonly display poor conservation across species [36], making them challenging to study using conventional biological techniques. Relative to their genomic location, lncRNAs are divided into five subclasses: sense (overlap exon on the same strand); antisense (located complementary to the sense strand); long intergenic (lincRNA) (located between coding genes, but their transcription occurs individually); bidirectional (located between coding genes and transcribed simultaneously); and intronic (produced by an intronic region) [37]. The schematic diagram of lncRNA biogenesis, types, and mechanism of gene regulation is presented in Figure 2.

## 4. LncRNA Databases

Several databases have been established to collect information regarding ncRNAs [41]. Databases such as LNCipedia [42], lncRNome [43], deepBase [44], and LncBook [45] are specially curated to categorize lncRNAs. However, they are limited to non-insect species, mainly humans and mice. NonCode, a more generalist database, comprises lncRNAs reported (experimental or computational approaches) from several model organisms, including *D. melanogaster* [46]. BmncRNAdb offers extensive information on the lncRNAs identified in the silkworms by utilizing RNA-seq and unigenes [47]. Similarly, InsectBase 2.0, a recently created database dedicated to insects, has curated several genomes and reported 1,293,430 lncRNAs from 376 insect species using computational tools such as FEELnc (v.0.2) [48,49]. Some of the available databases for lncRNA studies are listed in Table 1.

## 5. Functional Evidence of lncRNAs in Insects

In the last decade, researchers have identified some lncRNAs from insect species; however, nearly all developments in defining the functions of lncRNAs are focused on plants and mammals. *D. melanogaster* was the first insect species to have lncRNA identified from RNA-seq data [18]. Subsequently, researchers focused on various other insect species, such as *Apis cerana* and *Apis mellifera* [55], *Nilaparvata lugens* [56], *Plutella xylostella* [57], *Pectinophora gossypiella* [58] *Tribolium castaneum* [59] *Anopheles gambiae* [60], *Bombyx mori* [61], *Aedes aegypti* [62], *Sogatella furcifera* [63], and *Bactrocera dorsalis* [64].

LncRNAs are poorly characterized in insects, and most of our understanding comes from RNA-seq analysis. However, recent studies on model insect species, i.e., *Drosophila,* have identified lncRNAs involved in the development, defense mechanism, and resistance. In the subsequent segments, we will review the progress and mechanisms of lncRNAs in different insect species. The lncRNAs identified and functionally characterized in insect arthropods are presented in Table 2.

### 5.1. Drosophila melanogaster

*D. melanogaster* is an established model insect species for studying disciplines from fundamental genetics to deciphering complex genetic mechanisms. LncRNAs have been attributed to a variety of biological roles in *D. melanogaster,* including immunity [79], nervous system [93], spermatogenesis [94,95], and dosage compensation [66]. We will individually discuss the lncRNAs that have been studied in *Drosophila*.

#### 5.1.1. LncRNA-RNA-on-X (*roX*)

Dosage compensation is an epigenetic mechanism by which an organism equalizes the gene expressions in both sexes of a species. Organisms have evolved specialized molecular solutions to perform this task. In *Drosophila*, a dosage compensation complex (DCC) is formed by the two lncRNAs, *roX1* and *roX2,* with five male-specific lethal (MSL) proteins (MLE, MSL1/2/3, and MFO) and up-regulate genes on the single male X chromosome by facilitating the acetylation of histone at lysine 16 (H4K16ac), subsequently increasing gene transcription to balance the expression of two X chromosomes in females [65,66]. MSL1 binds to other proteins, including MSL3, which comprises of chromobarrel and binds to RNA [96]. MSL1/2 recognizes the binding sites, known as high-affinity sites, over the X chromosome, and MOF triggers the H4K16ac, which spreads this chromatin modification [65]. A schematic diagram of this mechanism is presented in Figure 3. It is worth mentioning that both of these lncRNAs function independently. The absence of *roX1* diminishes the expression of X-chromosome genes and the loss of *roX2* results in MSL-independent gene expression [97].

Contrary to *Xist*, which predominantly spreads and functions in *cis* of X chromosomes in mammalian females, *roX1* and *roX2* in *Drosophila* possess a unique feature and act in *trans*, indicating the functional versatility of lncRNAs in the modulation of a single chromosome in the two different systems [98]. Since sex determination and dosage compensation have co-evolved, scientists have recognized sex-lethal (*Sxl*) as the gene responsible for determining gender in *Drosophila* [99]. Sxl_Pe_, a dose-sensitive promotor, can distinguish between X chromosomes (1 vs. 2), which are only transcribed in females. Because sex determination and dosage compensation are connected, the lncRNAs may have a potential role in promotor activation and, eventually, gender decision [100].

**Figure 3 ijms-24-02605-f003:**
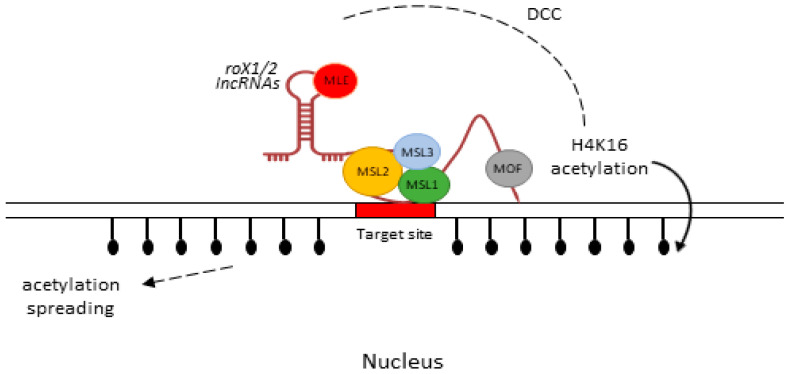
Regulatory role of *roX1/2* lncRNAs on the X chromosome of male *Drosophila*. Two lncRNAs and five male-specific lethal (MSL) proteins form a dosage compensation complex (DCC) [66]. The DCC is established and extends to the target site via diffusion. MLE is responsible for RNA/DNA helicase and ATPase activity [101]. MSL1/MSL2 identify the target site and MOF initiates the H4K16ac, spreading this chromatin modification over numerous kilobases [65].

#### 5.1.2. LncRNA-Heat Shock RNA Omega (*hsrω*)

The lncRNA *heat shock RNA omega* (*hsrω*) is a stress-induced and developmentally expressed lncRNA. Identified from the single-cell transcriptome atlas of the *Drosophila* brain, the lncRNA *hsrω* is involved in cellular aging [68]. The lncRNA *hsrω* (previously termed *93D*) is one of the earliest documented lncRNA genes [69], and its multiple transcripts (nuclear and cytoplasmic) are expressed in approximately all cells during development. Under stressful conditions, including heat shock stress, *hsrω* is critical for normal development [70]. Various species of *Drosophila,* i.e., *D. melanogaster*, *D. hydei,* and *D. pseudoobscura,* were compared for the structural analysis of lncRNA *hsrω*, which showed common primary structural characteristics with one intron and two exons. The gene produces three splice variants, with one cytoplasmic (hsrω-c) and the remaining two confined to the nucleus (hsrω-pre-c and hsrω-n) with non-significant ORF [102,103]. The nucleoplasmic transcript (hsrω-n) accumulates and restores vital regulatory factors (i.e., hnRNPs, RNAP II) to an optimum level (pre-stress) in cells that are recovering from heat stress, failure of which can be fatal [104,105]. Under thermal pressure, the silencing/overexpression of *hsrω* can be lethal to the embryonic and larval stages [106]. These findings highlight the crucial regulatory function of lncRNA *hsrω* in *Drosophila* cellular reprogramming, which is essential for survival during thermal stress.

#### 5.1.3. LncRNA-*yar*

The sleeping pattern in *Drosophila* is regulated by a conserved lincRNA, termed *yellow-achaete* intergenic RNA (*yar*), and expressed through mid-embryogenesis. Conserved across *Drosophila* species, *yar* represents an evolution of over 50 million years. [67,107]. Located in the neural gene cluster, the upstream of *yar* is the *yellow* gene (y) which codes a protein essential for cuticular pigmentation and sexual performance in males [108,109,110,111], whereas the downstream is the *achaete* gene (ac), encoding one of four bHLH (basic helix–loop–helix) transcription factors of theachaete–scute complex (AS-C), crucial for the development of the central and peripheral nervous system [112,113,114,115]. The functional studies revealed that nullisomy of *yar* did not affect flies morphologically; however, it caused deprived and fragmented nighttime sleep, with *yar* mutants showing reduced rebound sleep followed by sleep deficiency, indicating a critical role of *yar* in sleep homeostasis. Since *yar* is a cytoplasmic lincRNA and no evidence advocates its role in the transcription of neighboring genes, a potential relationship between *yar* and miRNA was studied. The subsequent investigation identified a total of 33 miRNAs. Seed matching confirmed 19 miRNAs in the exon of *yar*, suggesting that *yar* may function as a miRNA sponge [67]. Notably, among the matched miRNA seeds, one corresponds to the miR-310 cluster, and studies revealed that the lack of miRNAs 310–313 disrupts synaptic transmission at the neuromuscular junction [116]. These findings demonstrated the potential function of *yar* in the transcriptional circuit that affects miRNAs in nervous and, ultimately, sleep homeostasis. 

#### 5.1.4. LncRNA-*iab-8*

In *Drosophila*, a bithorax complex (BX-C) is in charge of the determination of abdominal sections and posterior thorax via regulation of three homeotic genes, i.e., *Ultrabithorax* (*Ubx*), *abdominal-A* (*abd-A*), and *Abdominal-B* (*Abd-B*) [117]. Several ncRNAs are concurrently expressed from the BX-C to *cis*-modulate specific proteins in abdominal segments [118,119]. The lncRNA *iab-8* (∼92kb) is spliced, polyadenylated, and encoded from the intergenic region flanked by the *abd-A* and *Abd-B* genes. It is expressed in nerve cells from the eighth abdominal segment and regulates the expression of *abd-A* via two mechanisms: (1) by acting as a precursor for miRNA, which results in the production of miR-iab-8 embedded within the intronic region of lncRNA *iab-8*, and (2) by regulating transcriptional interference (TI) of the *abd-A* promotor in a *cis* action [71]. The target sites of miR-iab-8 reside in the homeotic genes and their cofactors (*hth* and *exd*) [120,121]. A schematic diagram of this mechanism is presented in Figure 4. The knockdown of lncRNA *iab-8* can lead to sterility, mainly due to behavioral phenotype changes. In male flies, the inability to bend the abdomen prevents them from mating, whereas in females, it potentially causes peristaltic wave disorder, preventing the oviduct from facilitating the eggs’ movement [71].

#### 5.1.5. LncRNA-*oskar*

Scientists have discovered that some protein-coding genes can potentially function as lncRNAs [122]. In humans, several mRNAs, such as *insulin receptor substrate 1* (IRS1) and *p53*, can function as regulatory RNA [123,124]. Similarly, a maternal gene *oskar* in *Drosophila* is essential for posterior abdominal segmentation and germline determination [72]. During the initial stages of oogenesis, *oskar* RNA plays a translation-independent role and acts as a lncRNA. Reduction in *oskar* levels results in a sterile phenotype chiefly due to oogenesis arrest. However, the expression of *oskar* 3’UTR is adequate for the recovery of egg-less defects of the RNA null mutant independent of protein [73]. Staufen, an RNA-binding protein, is localized inside oocytes in association with the *oskar* RNA [125]. In *oskar* null mutants, the Staufen protein fails to transport from the nurse cells into the oocyte, revealing the interdependency of Staufen and *oskar* RNA regulated via the interaction of Staufen with *oskar* 3’UTR [73]. However, this non-coding function may probably be mediated via the sequestration of Bruno, a translation regulator which binds to the Bruno responsive element in 3’UTR [122].

#### 5.1.6. LncRNA-*CR46018*

The Toll pathway is an integral part of insect immunity that recognizes invading pathogens and induces antimicrobial peptides (AMPs) to neutralize the microorganisms [126]. In mammals, numerous lncRNAs, e.g., *Lethe* [127], *lincRNA-Cox2* [128], *THRIL* [129], and *PACER* [130], represent the growing list of lncRNAs involved in regulating gene expression in the innate immune system. However, the implication of lncRNAs in regulating insect immune responses is yet to be studied. In *Drosophila,* the infection of *Micrococcus luteus* induced the expression of lncRNA-*CR46018,* suggesting its potential role in the immune response. RNA-seq analysis of the lncRNA-*CR46018* mutant flies showed that the overexpression of *CR46018* results in the up-regulation of several immune-related genes, mainly in the Toll and IMD pathways. Additional experiments revealed that lncRNA-*CR46018* interacted with transcription factors Dif and Dorsal in the Toll pathway. When infected with *M. luteus*, the *CR46018* overexpressing flies showed promising survival abilities compared to the control [74]. The same group studied the function of another novel lncRNA called lncRNA-*CR11538*. The results showed that lncRNA-*CR11538* acts as a Dif/Dorsal decoy resulting in the down-regulation of AMPs and restoration of *Drosophila* Toll immunity homeostasis [75]. The above research highlights the regulatory function of lncRNAs in *Drosophila* Toll immunity.

### 5.2. Apis mellifera

*A. mellifera* is among the most widely studied insects owing to its environmental and economic potential. However, the studies concerning the role of lncRNAs are insufficient, and only a few have been investigated functionally. Most of the lncRNAs studied are involved in neuronal functions [83]. Other lncRNAs, such as *Lncov1/2* (intronic), have been reported from queen ovaries and are responsible for juvenile hormone (JH) dependent maintenance of ovarioles [131]. Similarly, lncRNAs *TCONS_00356023*, *TCONS_00357367*, and *TCONS_00159909* have also been identified for their potential roles in behavioral transition [132].

#### 5.2.1. LncRNA-*Nb-1*

Honey bees are eusocial insect species that represent a highly synchronized working of a society. The adult females are divided into two castes, queen (reproduction) and workers (labor). The honey bees have an age-dependent work division, with young workers responsible for brood care. In contrast, elderly workers are responsible for foraging pollen and nectar [133]. Studies have shown that the nurse-forager transition is mediated by a plethora of carefully coordinated interactions of hormonal levels [134], colony demography [135], gene expressional changes in the brain [136], exocrine gland activity, brain chemistry and structure [137,138]. Studies have been conducted to detect the possible involvement of lncRNAs in the nursing-foraging transition in *A. mellifera* [132]. LncRNA-*Nb-1* (Nurse bee brain-selective gene-1), a 700 nt transcript, appears to function in age-dependent transition by modulating the production and secretion of octopamine and JH. In comparison, the expression of lncRNA-*Nb-1* was more significant in the brains of nursing bees relative to foraging and queen bees [81], implying its critical role in the behavioral transition.

#### 5.2.2. LncRNA-*Ks-1*

In the western honey bee *A. mellifera*, a 17kb nuclear lncRNA-*Ks-1* (Kenyon cell/small-type preferential gene-1) is specifically expressed in the mushroom body (MB) of Kenyon cells in the brain. The Kenyon cells play critical roles in the regulatory sections of the insect brain. The lncRNA-*Ks-1* is chiefly expressed in the brain of drones compared to queen bees and is present between the brain’s optic lobes and lateral calyx, signifying its crucial role in drone-related functions [82]. This study identified another transcript termed LncRNA-*AncR-1*, mainly expressed in the sexual tissues, secretory organs, and the brain [83]. Both these transcripts are primarily present in the neuronal nuclei, suggesting their potential neural functions.

#### 5.2.3. LncRNA-*kakusei*

In *A. mellifera*, a nuclear lncRNA, termed *kakusei* (∼7 kb long), was identified and presented the neuronal activity pattern in foragers [139]. The lncRNA-*kakusei* comprises various constitutive and inducible variants, and its expression is briefly up-regulated during neuronic activities and is essential to multiple neural functions and RNA metabolism. 

Additionally, a homolog of *kakusei* was also detected and isolated from active neurons of worker bees in *A. cerana* and called Acks (*A. cerana kakusei*). It is predominantly expressed in worker bees while forming a hot defensive ball to counter a giant hornet attack [85]. These discoveries highlighted the neuronal function of lncRNAs in honey bees; however, a comprehensive investigation is required to elucidate the underlying mechanisms.

### 5.3. Bombyx mori

*B. mori* is among the widely known insects, owing to its economic (silk production) and scientific (model for lepidopteran studies) significance [140]. Over the years, it has continuously generated interest from the scientific community to investigate several known and novel scientific phenomena, including lncRNAs [61,141,142]. We will discuss the progress of lncRNAs in silkworms in the following section.

#### 5.3.1. LncRNA-*Bmdsx-AS1*

Insects exhibit a series of sex-determination mechanisms, including the composition of sex chromosomes, environmental factors, and pathogen manipulation [143,144]. An antisense lncRNA-*Bmdsx-AS1* is abundantly expressed in the *B. mori* testicle. It functions in the sex-specific alternative splicing of its *doublesex* (*Bmdsx*) gene [86]. In insects, the *dsx* is alternatively spliced into several transcripts, eventually producing sex-specific proteins responsible for sexual dimorphism [145]. The lncRNA-*Bmdsx-AS1* binds specifically to the splicing factor hnRNPH and interacts with *B. mori* P element somatic inhibitor (BmPSI) [86], a male-specific protein responsible for splicing of *Bmdsx* pre-mRNA [146]. Subsequent studies were performed in transgenic *Bmdsx-AS1* to investigate changes in male genitalia. The overexpression of lncRNA-*Bmdsx-AS1* altered the morphological structure of male genitalia by increasing the number of claspers compared to the wild-type. Additionally, the overexpression of *Bmdsx-AS1* reduced the expression of genes in the EGFR (Epidermal Growth Factor Receptor) signaling pathway and vice versa [87]. EGFR is crucial for developing the eighth abdominal segment in *B. mori* [147]. Promotor analysis demonstrated that *BmAbd-B* (gene related to exterior genital development) can negatively modulate the expression of *Bmdsx-AS1* [87]. These studies presented a multilayered regulatory network involving *BmAbd-B*, *Bmdsx,* and *Bmdsx-AS1*.

#### 5.3.2. LncRNA-*Fben-1*

The silkworm *B. mori* displays sexually dimorphic behavior [148]. The female releases pheromones by extending her pheromones glands to attract mates, whereas the males exhibit typical sexual behavior in response to sex pheromones [149]. Molecular studies of this sexually dimorphic behavior suggested the potential role of a nuclear lncRNA, the lncRNA-*Fben-1* (female-brain expressed non-coding RNA-1). The lncRNA-*Fben-1* is one of the earliest known lncRNA identified in silkworms. In the *B. mori* genome, the lncRNA-*Fben-1* is located ~6 kb upstream of the fruitless (*fru*) gene. The expression of lncRNA-*Fben-1* is preferential to the cells around the mushroom bodies of the adult female brain, signifying its crucial role in sexually dimorphic neuronal functions in females [88].

#### 5.3.3. LncRNA-*209997*

The *B. mori* nucleopolyhedrovirus virus (BmNPV) is an important pathogen. The pathogen severely infects silkworms and seriously damages the sericulture industry [150]. Transcriptomic investigations revealed the role of lncRNAs in *B. mori* immune responses during baculovirus infection [151,152]. Viral infections can manipulate the expression of lncRNAs, and these lncRNA transcripts can either promote or inhibit viral replication [153,154]. Cellular lncRNA, termed *Lnc_209997,* was found in silkworm’s virus-infected fat body tissues. The expression of *Lnc_209997* was significantly downregulated during infection, thus promoting virus replication. However, the overexpression of *Lnc_209997* inhibited viral replication, suggesting its critical role during infection [89]. Researchers have documented that lncRNAs can bind to miRNAs as competitive endogenous RNAs (ceRNAs) and control their target genes, ultimately affecting viral replication [155]. A similar interaction was observed between *Lnc_209997* and miR-275-5p during BmNPV infection. Overexpression of *Lnc_209997* resulted in the inhibition of miR-275-5p and vice versa. These results suggest that BmNPV can potentially promote its proliferation by suppressing *Lnc_209997*, which diminished the relationship between *Lnc_209997* and miR-275-5p and increased the production of miR-275-5p, therefore promoting its replication [89]. Notably, this hypothesis was not experimentally validated and needs further evidence to decipher the molecular mechanism involved.

### 5.4. Plutella xylostella

*P. xylostella* is one of the brassica crop’s most destructive lepidopteran pests [156]. The insect is notorious for resisting multiple insecticides, such as *Bacillus thuringiensis* (Bt), Fipronil, and Chlorpyrifos. Researchers have identified several lncRNAs for their potential role in developing resistance. Many of these lncRNAs may act as precursors for the production of miRNAs [157]. The glutathione S-transferase (GST) is a crucial detoxification enzyme that contributes to resistance by aiding metabolism or chemical sequestration, resulting in cell excretion [158]. An antisense lncRNA, lnc-*GSTu1-AS* was identified from chlorantraniliprole-resistant populations of *P. xylostella*. The lncRNA is transcribed from the opposing strand of *GSTu1*, a gene associated with chlorantraniliprole resistance. Studies have shown that the knockdown of *GSTu1* significantly reduced the *P. xylostella* resistance to chlorantraniliprole. Furthermore, miR-8525-5p was identified to be regulating the activity of *GSTu1*. It is worth mentioning that in vivo suppression/overexpression did not change the expression of *GSTu1*, signifying that an additional layer of gene regulation may be present via lncRNAs. Subsequent experimentations demonstrated that the lnc-*GSTu1-AS* steadied the *GSTu1* expression by blocking the binding site of miR-8525-5p at the 3’UTR of *GSTu1*, thus preventing its degradation and ultimately increasing resistance to chlorantraniliprole [91]. LncRNAs are expressed differentially under various biotic and abiotic stresses [159]. In response to *Metarhizum anisopliae* infection, multiple lncRNAs were identified, regulating mRNA expression and acting as miRNA precursors. Bioinformatic analysis predicted the function of these lncRNAs in regulating many immune-related genes, including *βGRP*, *toll*, *serpin* and *transferrin* [160]. Several differentially expressed transcripts were identified from dsRNA-induced *P. xylostella* libraries. *Dicer-2* and the lncRNA targeting it were significantly expressed post-dsRNA treatment, indicating the crucial roles lncRNAs in the modulation of RNAi pathways [161].

### 5.5. Aedes aegypti and Aedes albopictus

*Ae. albopictus* and *Ae. aegypti* are the primary vectors of chikungunya, dengue fever, yellow fever, and Zika virus [162]. Researchers have investigated the critical modulatory role of host lncRNAs during virus interactions [163]. Recent studies identified and functionally annotated lncRNAs in *Ae. aegypti* [62,164]. The significance of lncRNAs was investigated in dengue virus serotype 2 (DENV-2) and *Wolbachia*-infected groups. The results showed that the DENV-2 infection amplified the profusion of lncRNAs in mosquito cells, with some able to suppress virus replication. The silencing of *lincRNA_1317* promoted viral replication, suggesting its role in mosquito defense mechanisms against viruses.

Similarly, *Wolbachia* infection led to several differently expressed lncRNAs [62,165]. Subsequent studies revealed that *Wolbachia* employs the host lncRNA to maintain intracellular oxidative stress and initiate immune responses. The lncRNA, *aae-lnc-7598,* was significantly induced during *Wolbachia* infection and modulated the reactive oxygen species (ROS) via the *trans*-regulation of an antioxidant gene, *CAT1B*. Furthermore, *Wolbachia* reduced the expression levels of another lncRNA, *aae-lnc-0165*. The suppression of *aae-lnc-0165* induced the expression of *REL1*, a gene crucial for downstream activation of the Toll pathway, by sequence-specific binding of aae-miR-980-5p. Notably, suppressing *aae-lnc-0165* also reduced cell ROS levels [92]. Transcriptional analysis performed in chikungunya infected midgut of *Ae. albopictus* identified several differentially expressed lncRNA transcripts, likely in response to infection [166]. Besides their role in vector–virus interaction, lncRNAs are crucial players during development. RNA-seq data from different development stages in *Ae. albopictus* revealed that most of the lncRNAs are up-regulated at the start of metamorphosis, whereas the egg and the early larval stage had a greater number of differentially expressed transcripts, signifying their highly spatiotemporal expression patterns [167].

### 5.6. Other Insects

LncRNAs have become a global research hotspot, generating increased interest from various entomological research groups. Even though most of the work is conducted on the species mentioned above, RNA-seq has enabled us to identify lncRNAs from several other insect species. In the malarial vector *A. gambiae*, 2949 lncRNAs were identified from different life stages, most being intergenic [168]. *S. furcifera*, a destructive pest of rice, was profiled for lncRNAs during different developmental stages and identified 1861 lncRNAs. Most of the lncRNAs were up-regulated in embryo, fourth and fifth instars, indicating the potential role of lncRNAs during development [63]. In order to address the problem related to Bt-resistance in bollworm *Helicoverpa zea*, a comparative study was conducted between Cry1Ac-resistant and Cry1Ac-susceptible strains. Results showed that multiple lncRNAs were differentially expressed in the resistant strains. Further analysis showed a pseudogenic lncRNA similar to cadherin. Additionally, these lncRNAs showed their proximity to protein-coding genes, including ABC transporter and serine protease, possibly crucial to the mechanism of resistance development [169]. 

Similarly, in pink bollworm, *P. gossypiella,* the lncRNA *PgCad1* was identified. It regulates *P. gossypiella* cadherin (*PgCad1*), a gene involved in resistance development. Knockdown of *lncRNA PgCad1* suppressed the transcription of *PgCad1* and reduced the susceptibility to Cry1Ac, suggesting that this lncRNA positively regulates cadherin’s expression and increases the vulnerability of pink bollworm larvae [58]. Whole-transcriptome RNA-seq, performed in two strains of *Aphis gossypii* (spirotetramat-resistant and susceptible), a notorious sap-sucking polyphagous pest of crops. Differential expression analysis revealed 874 lncRNAs, with five lncRNAs predicted as acetyl-CoA carboxylase (ACC) targets. RT-qPCR combined with RNAi confirmed the interaction of lncRNAs with the expression of ACC genes. Additionally, two transcription factors, C/EBPzeta and C/EBP, were found to regulate the transcription level of lncRNAs [170]. The study presented valuable information for understanding the function of *A. gossypii* lncRNAs in the development of resistance. In *B. dorsalis*, malathion-resistant and susceptible strains were subjected to RNA-seq, resulting in the identification of 6171 lncRNAs with multiple differentially expressed transcripts. RT-qPCR analysis identified lncRNAs associated with malathion resistance, with *lnc3774.2* and *lnc15010.10* significantly expressed in the cuticle of the malathion-resistant population, signifying their roles in resistance. Suppression of *lnc3774.2* indicated that cuticular changes could be critical in malathion resistance [64]. The testis-specific lncRNA, *lnc94638,* was identified in the melon fly, *Zeugodacus cucurbitae*, an extremely destructive insect pest of cucurbit crops. Localized in the apical testicle region containing mature spermatozoa, the knockdown of *lnc94638* drastically lowered the number of spermatozoa and affected male fertility [171]. Genome-wide studies of different developmental stages in *T. castaneum* demonstrated that lncRNAs are crucial for metabolism and cell differentiation [59].

## 6. Research Challenges and Future

Research on lncRNAs has gained momentum in many plant and animal groups, but insects are still poorly understood. Studies conducted among insects are limited to a few model species, mainly in *D. melanogaster,* and our knowledge of lncRNAs from other insect species is still in its primitive stage. Since lncRNAs interact with several genetic molecules, i.e., protein, DNA, and RNA, and subsequently modulate gene expression at the epigenetic, transcriptional, and post-transcriptional levels, studying gene expression could help us understand their biological phenomena. An in-depth elucidation of the sequence and structural elements related to lncRNA function will permit us to predict and classify lncRNAs into families, as has happened for proteins with similar structural domains.

Technological advances such as RNA-seq and bioinformatic tools enabled us to detect and characterize lncRNAs to a certain extent. However, due to a lack of available functional information, filtering the lncRNAs from protein-coding RNAs is still challenging. Recently we have seen the reclassification of former ncRNAs (e.g., tarsal-less/polished rice and pgc) as coding for small peptides in *Drosophila* and *Tribolium* spp. [172]. The development of devoted bioinformatic tools is required to identify multifunctional RNAs. High-resolution in vivo RNA imaging, comprehensive identification of RNA-binding proteins (ChIRP), and capture hybridization analysis of RNA targets (CHART) are some of the modern experimental methods that can be employed to identify the protein, RNA, and DNA binding partners of lncRNAs. However, these experiments must be performed across various species to determine the success ratios. Computational approaches, including the construction of designated databases and the compilation of available annotation resources, are required to collect and integrate information about lncRNAs across various species.

Another direction is understanding how lncRNAs cross-talk with each other and how these peculiar cross-talks are modulated. The spatiotemporal transcriptome can provide a comprehensive resource for understanding the function and interaction of lncRNAs in different tissues and cells at a particular time. Similar studies in *Drosophila* identified a set of conserved lncRNAs, regulated in a tissue-specific pattern, and revealed a complex spatiotemporal expression of lncRNAs during neurogenesis [173]. Novel sequencing technologies that directly sequence RNA and RNA modifications without the need for reverse transcription could lay the ground for further studies. However, developing advanced genetic model systems is essential to understand lncRNA functions in vivo and in vitro.

## 7. Conclusions

Compiling evidence suggests the critical function of lncRNAs in correctly executing gene expression. The examples reviewed and discussed here demonstrate the functional versatility of lncRNAs in various species of insects. Despite the number of proposed models, only a fraction lncRNAs have been ascribed to a specific target and named. For most insect lncRNAs, information comes only from RNA-seq experiments. As mentioned above, technological advances have enabled researchers to obtain results for thousands of lncRNA transcripts in a single investigation. The focus of biological research has shifted towards a holistic approach with the integration of systematic biology, which gives us a comprehensive interpretation of the lncRNA data via computational modeling. However, there is a clear need to develop experimental techniques that would allow us to differentiate lncRNAs from mRNAs and to understand the interaction of lncRNAs with other genetic molecules. Hopefully, the current flurry of attention on lncRNAs will be critical for resolving these complications from widespread experimental discoveries concerning the structural and functional mechanism of lncRNAs in insects.

## Figures and Tables

**Figure 1 ijms-24-02605-f001:**
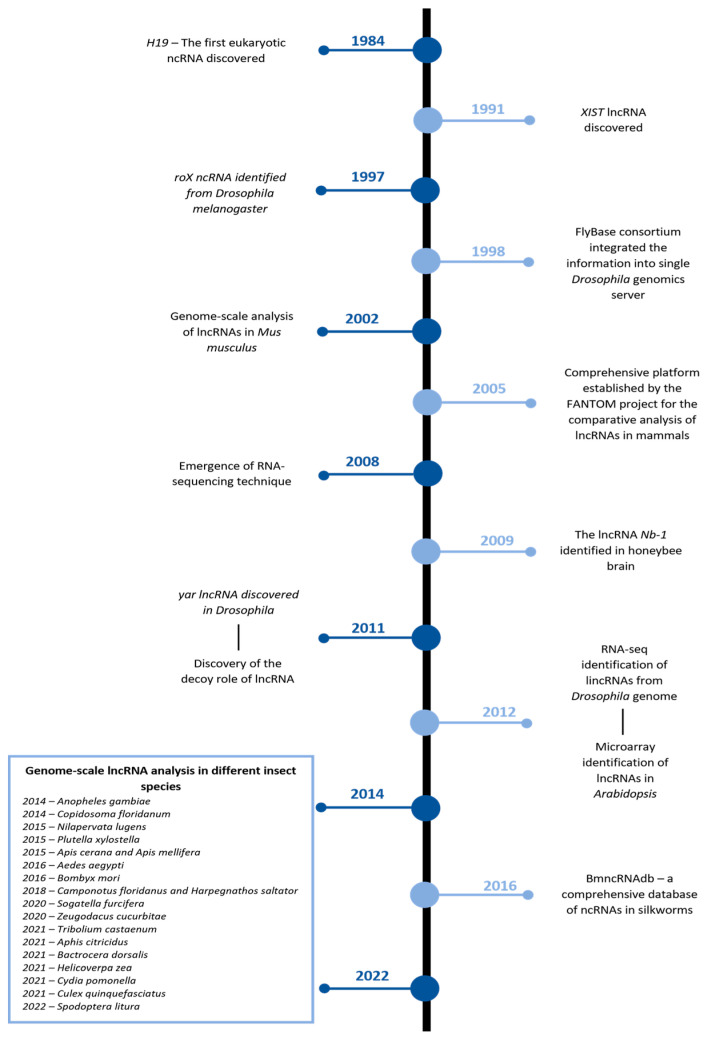
Some of the significant developments in long non-coding RNA research. Note: lncRNA denotes long non-coding RNA; *roX* denotes RNA-on-X; *XIST* denotes X-inactive-specific transcript; *Nb-1* denotes Nurse bee brain-selective gene-1; yar denotes yellow-achaete intergenic RNA; and lincRNA denotes long intergenic non-coding RNAs.

**Figure 2 ijms-24-02605-f002:**
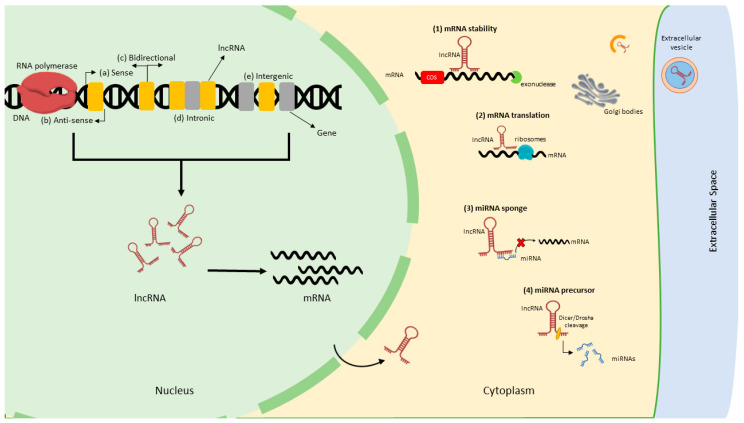
Biogenesis, classification, and mechanism of gene regulation by lncRNAs. LncRNA is transcribed by RNA polymerase, which can be categorized into several types: (a) sense; (b) antisense; (c) bidirectional; (d) intronic, and (e) intergenic [37]. The mechanism of gene regulation of lncRNAs is presented: (1) mRNA stability (via recruitment or impairment of RNA degrading enzymes such as exonucleases) [38], (2) mRNA translation (via recruitment or impairment of ribosomes) [39], (3) miRNA sponge (lncRNAs can prevent miRNA by targeting mRNAs by acting as competing endogenous RNAs (ceRNAs) [31], and (4) miRNA precursors (lncRNAs harbor miRNAs by acting as precursors) [40].

**Figure 4 ijms-24-02605-f004:**
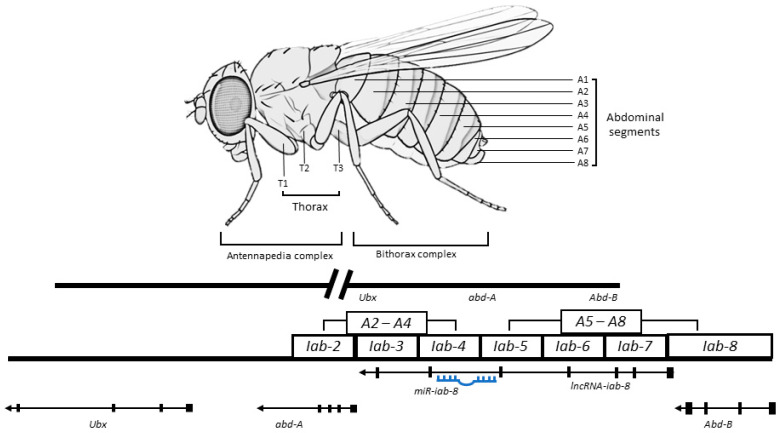
Schematic diagram of the bithorax complex (BX-C) and the lncRNA-*iab-8*. The segment-specific domains are presented in rectangles. The lncRNA-*iab-8* encodes from the intergenic region between the *abd-A* and *Abd-B* genes. It is expressed in neural cells from the eighth abdominal segment and regulates the expression of *abd-A*. The lncRNA-*iab-8* regulates via acting as a precursor for miR-iab-8 production or by controlling transcriptional interference of the *abd-A* promotor in a *cis* manner [71] Note: Figure adapted with permission from Daniel L. Garaulet, Eric C. Lai, Mechanisms of Development; published by Elsevier, 2015. Copyright license no. 5462280266230.

**Table 1 ijms-24-02605-t001:** List of some crucial databases available for lncRNA studies.

Databases	Features	Links	References
InsectBase 2.0	The database contains more than 16 million sequences, including 817 insect genomes in 20 orders, 158 families, and 457 genera. It includes 1,293,430 lncRNAs from 376 species.	http://v2.insect-genome.com (accessed on 15 November 2022)	[49]
BmncRNAdb	The database offers extensive information on the lncRNAs identified in the silkworms. It compiled a list of 6281 lncRNAs with the help of RNA-seq data and unigenes.	http://gene.cqu.edu.cn/BmncRNAdb (accessed on 30 November 2022)	[47]
FlyBase	A dedicated repository of *Drosophila* genes and genomes.	http://flybase.org (accessed on 15 November 2022)	[50]
LncRBase V.2	The database contains information about lncRNAs from eight species, including *Drosophila*. It provides information on transcript features, genomic location, promoter information, and sORF information within lncRNAs.	http://dibresources.jcbose.ac (accessed on 10 November 2022)	[51]
RNAcentral v20	Multispecies database. RNAcentral imports ncRNA sequences from multiple databases. It has imported data from 47 databases, including Ensemble, LNCipedia, lncRNAdb, and NonCode.	https://rnacentral.org (accessed on 15 November 2022)	[52]
NonCode v6.0	Dedicated database for ncRNAs. Collection of 39 species, including *D. melanogaster*. It provides basic and advanced information on lncRNAs, such as expression profile, conservation info, and functional prediction.	http://www.noncode.org (accessed on 15 November 2022)	[46]
LNCediting	It offers a comprehensive resource for the functional prediction of novel RNA editing sites in lncRNAs amongst multiple species, including *Drosophila*.	http://bioinfo.life.hust.edu.cn/LNCediting (accessed on 10 November 2022)	[53]
CRISPRlnc	Manual of confirmed CRISPR/Cas9 sgRNAs for lncRNA.	http://www.crisprlnc.org/browse (accessed on 8 November 2022)	[54]

**Table 2 ijms-24-02605-t002:** List of lncRNAs identified from different insects.

LncRNAs	Functions	References
*Drosophila melanogaster*		
*roX1* and *roX2*	Dosage compensation	[65,66]
*yellow-achaete intergenic RNA (yar)*	Sleeping mechanism	[67]
*heat shock RNA omega (hsrω)*	Cellular aging	[68,69,70]
*iab 8*	Behavioral phenotype	[71]
*oskar*	Abdominal segmentation	[72,73]
*CR46018* and *CR11538*	Toll immunity	[74,75]
*CR33942*	Imd immune response	[76]
*IRAR*	Insulin signaling	[77]
*CRG*	Locomotive behavior	[78]
*IBIN*	Immunity and metabolism	[79]
*sphinx*	Male courtship behavior	[80]
*Apis mellifera*		
*Nb-1*	The behavioral transition from nursing to foraging	[81]
*Ks-1* and *AncR-1*	Neuronal functions	[82,83]
*Lncov1* and *Lncov2*	Cell death of ovarioles in worker embryogenesis.	[84]
*kakusei*	Neural functions in forager bees	[85]
*Bombyx mori*		
*Bmdsx-AS1*	Sexual behavior	[86,87]
*Fben-1*	Role in sexually dimorphic neural functions in females	[88]
*Lnc_209997*	BmNPV replication	[89]
*lncR17454*	Metamorphosis	[90]
*Plutella xylostella*		
*lnc-GSTu1-AS*	Chlorantraniliprole resistance	[91]
*Aedes aegypti*		
*lincRNA_1317*	Dengue virus replication	[62]
*aae-lnc-7598*	Antioxidant functions	[92]
*aae-lnc-0165*	Toll pathway	[92]

## Data Availability

Not applicable.

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
