# Peer review of "Recent Advances and Future Potential of Long Non-Coding RNAs in Insects"

_ijms, 2023, doi:10.3390/ijms24032605_

Round 1

Reviewer 1 Report

Authors Zafar J, Huang J, Xu X, and Jin F have done a great job in writing a consolidated report on lncRNAs across a variety of insects. As a majority of known lncRNAs are found to be enriched in a species and not conserved across taxa, study of their significance is truly challenging. Despite the challenge, this reviewer is grateful scientists continue this work. 

There are only a minor things that should be addressed. 

line 35, 'our discussion would' should be 'our discussion will'

lines 71-85, note that not all lncRNAs are poly-adenylated, also with newer references the authors should note a number of lncRNAs are found in cytoplasm over nucleaus

lines 345-362, names of miRNAs should be in italics. the authors put all other genes in italics so not sure why miRNAs are not in italics

Author Response

Dear,

Thank you

Reviewer 2 Report

The review paper”recent advances and future potential of long non-coding RNAs in insects” is of interest to the RNA research community. The review included two parts. One is listed the database, which focused on lncRNA or included lncRNA. The other part is overviewed the most important lncRNA so far discovered. In general, the information about lncRNA is very little, especially for lncRNA with functional characterization, which were still concentrated on the model insect species. I am wondering that if authors can add how many lncRNA estimated in table 1 for each insect species, that will be very helpful.  I see in the paragraph ”other insects” touched on these information. Maybe it require manually processing information. Because lncRNA is so new to all the authors, if that crucial information provided, it could set apart yours from other review. Is there any information in the literature about how lncRNA formed from evolutionary point?

Minor comments
line 114& 115, two titles for a table.

Line 478. Another direction is to understand how lncRNAs cross-talk with each other and how these peculiar ~~~

This paper from Drosophila also is very interesting, should cite for your review paper: A gene expression atlas of embryonic neurogenesis in Drosophila reveals complex spatiotemporal regulation of lncRNAs

Author Response

Dear,

Thank you...!
